# Scd1 Deficiency in Early Embryos Affects Blastocyst ICM Formation through RPs-Mdm2-p53 Pathway

**DOI:** 10.3390/ijms24021750

**Published:** 2023-01-16

**Authors:** Huimin Niu, Anmin Lei, Huibin Tian, Weiwei Yao, Ying Liu, Cong Li, Xuetong An, Xiaoying Chen, Zhifei Zhang, Jiao Wu, Min Yang, Jiangtao Huang, Fei Cheng, Jianqing Zhao, Jinlian Hua, Shimin Liu, Jun Luo

**Affiliations:** 1Shaanxi Key Laboratory of Molecular Biology for Agriculture, College of Animal Science and Technology, Northwest A&F University, Yangling 712100, China; 2Shaanxi Stem Cell Engineering and Technology Research Center, College of Veterinary Medicine, Northwest A&F University, Yangling 712100, China; 3UWA Institute of Agriculture, The University of Western Australia, Perth, WA 6018, Australia

**Keywords:** Scd1, RPs-Mdm2-p53, ICM, lipid droplet, embryo development

## Abstract

Embryos contain a large number of lipid droplets, and lipid metabolism is gradually activated during embryonic development to provide energy. However, the regulatory mechanisms remain to be investigated. Stearoyl-CoA desaturase 1 (Scd1) is a fatty acid desaturase gene that is mainly involved in intracellular monounsaturated fatty acid production, which takes part in many physiological processes. Analysis of transcripts at key stages of embryo development revealed that Scd1 was important and expressed at an increased level during the cleavage and blastocyst stages. Knockout *Scd1* gene by CRISPR/Cas9 from zygotes revealed a decrease in lipid droplets (LDs) and damage in the inner cell mass (ICM) formation of blastocyst. Comparative analysis of normal and knockout embryo transcripts showed a suppression of ribosome protein (RPs) genes, leading to the arrest of ribosome biogenesis at the 2-cell stage. Notably, the P53-related pathway was further activated at the blastocyst stage, which eventually caused embryonic development arrest and apoptosis. In summary, Scd1 helps in providing energy for embryonic development by regulating intra-embryonic lipid droplet formation. Moreover, deficiency activates the RPs-Mdm2-P53 pathway due to ribosomal stress and ultimately leads to embryonic development arrest. The present results suggested that *Scd1* gene is essential to maintain healthy development of embryos by regulating energy support.

## 1. Introduction

The embryo is the starting point of life, with unique pluripotency and a high proliferation rate. Early embryonic development in mammals is controlled by multiple factors, including genetic factors and environmental influences [1,2,3], and all animal zygotes undergo maternal zygotic transformation (MZT) [4], during which maternal message and protein are subsequently inactivated and the embryonic genome is programmatically activated [5]. During this stage, energy metabolism is relatively quiescent until the blastocyst is formed, when it becomes highly metabolic activities [6]. The formation of the blastocoel is energetically expensive. The high rate of cell cleavage is also accompanied by an increasing level of total biosynthesis and high demands for energy [6]. There has been a massive volume of literature proving that carbohydrates, particularly pyruvate, lactate, and amino acids, are the primary energy source for the development of embryos [6,7]. Lipids stored in oocytes also play a role during this stage [8].

Lipids appear as droplets (LDs) in oocytes and embryos and are hydrolyzed to free fatty acids in the presence of energy demand to provide substrates for β-oxidation and oxidative phosphorylation in mitochondria [9]. Mouse oocytes have fewer LDs in GV and MII stages and early embryonic stages, with an average size of 0.3 μm, while in 8-cell to blastocyst stage embryos, LDs become fewer and larger [10]. The distribution of LD probably reflects the metabolic state of oocytes or embryos. Bradley and Swann proposed that for healthy mammalian oocytes, a balance between pyruvate and fatty acid oxidation is essential to maintain a low level of reactive oxygen radical damage [8]. This might also be the case for embryos, because the zygote and cleavage stage embryos are considered as an extension of oocytes in physiology [6]. Furthermore, the expression of lipid metabolism-related genes changed significantly during the MZT process [11]. However, the mechanism of fatty acid metabolism in early-stage embryo development is unknown.

Among lipid metabolism genes, stearoyl-CoA desaturase-1 (Δ-9-desaturase) (Scd1) is a rate-limiting enzyme that catalyzes the formation of monounsaturated fatty acids (MUFAs) from stearoyl-CoA and palmitoyl-CoA. MUFAs can serve as substrates for the synthesis of various kinds of lipids, including phospholipids, triglycerides, and also be used as mediators in signal transduction and differentiation [12,13]. There is little information about the effect of Scd1 on early embryo development. However, there are a number of reports on the effects of Scd1 in tumor or cancer cells [14,15,16]. To support proliferation, cancer cells need a large amount of saturated and monounsaturated fatty acids (SFA, MUFA), which are used for energy storage and production, membrane biosynthesis, and signaling [17,18,19]. Stearoyl-CoA and palmitoyl-CoA, on the other hand, can be oxidized via β-oxidation to provide energy for cell growth [13]. In contrast, inhibition of Scd1 expression induces ferroptosis and apoptosis in ovary cancer cells [20,21,22,23,24]. Scd1 may have similar effects on lipogenesis in embryonic development.

Cells with stem cell-like features exhibit a distinctive fatty acid metabolic profile marked by increased lipogenesis, especially the activity of stearoyl Scd1. In mouse induced pluripotent stem cells (miPSCs) and human embryonic stem cells (hESCs), Scd1 supports cell growth and survival [25]. Inhibition of Scd1 in vitro results in the specific eradication of pluripotent cells [26]. In ovaries, cumulus cells with high Scd1 expression have lower mortality, and oocytes have a higher maturation rate [27]. In human-induced pluripotent stem cells, Scd1 correlates with disorders of the endoderm-derived organs [28]. However, the regulatory relationships between Scd1 and embryo development needs further investigations [29]. Based on these results, we hypothesized that Scd1-related lipid metabolism could play an important role in early embryonic development. To test this hypothesis, the CRISPR/Cas9 technique was used to produce the *Scd1* knockout mouse embryos for investigating the functions of Scd1 during the embryonic development.

## 2. Results

### 2.1. Expression of Scd1 Gradually Increased as Embryonic Development

According to GSE1749 [30], there were 673 genes upregulated in 2-cell stage embryos, and 312 genes upregulated in the blastocyst stage. Combined with GSE18290 [31] in blastocyst stage (Figure 1A), there were four genes increased in both 2-cell and blastocyst stage embryos, indicating that these genes were significant for embryo development (Figure 1A,B). Amon them, *Scd1* gene was closely related to energy metabolism and provided energy for embryos. Following these results, we further designed the sgRNAs based on the sequence of the mouse *Scd1* gene in NCBI. The sgRNAs were designed on the CHOPCHOP website (http://chopchop.cbu.uib.no, accessed on 18 July 2020) to target the first and second Exon region, and 2 high-efficiency sgRNAs were selected [32] (Figure 1C, and Table 1) for embryo microinjection (Figure 1D and Appendix A). We first asked whether the concentration of sgRNA/Cas9 had an effect on the gene editing efficiency. To do so, we compared two different concentrations of sgRNA/Cas9 (25 ng μL^−1^/50 ng μL^−1^ or 50 ng μL^−1^/100 ng μL^−1^) (Appendix A) in mouse zygotes, and we found that the concentrations of 50 ng μL^−1^/100 ng μL^−1^ had a higher knockout efficiency of *Scd1*, reaching 89.58% and 80.55% for sgRNA3 and sgRNA6, respectively (Figure 1E). As a result, screening the key genes of embryonic development and efficient sgRNAs was critical for the creation of gene knockout embryos.

### 2.2. Scd1 Deficiency Caused Embryonic Development Arrest

To produce Scd1-knockout mice, we transplanted 1-cell stage embryos into the oviducts of pseudopregnancy mice after microinjection. A total of 480 NC sgRNA injected (NC ^sgRNA^) and 700 Scd1 sgRNA injected (Scd1^sgRNA^) zygotes were transplanted, 20 embryos for each recipient (Figure 2A). After 21 days of gestation, the NC ^sgRNA^ transplanted mice birthed 7 litters of 83 pups with the embryo loss rate of 40.7% ((1 − 83/140) ×100%), while Scd1^sgRNA^ transplanted mice birthed 10 litters of 38 mice with the embryo loss rate of 81.0%, significantly higher than that of the NC ^sgRNA^ transplanted mice (Figure 2I). The litter size of Scd1^sgRNA^ transplanted mice (3.8/Litter) was significantly lower than the NC^sgRNA^ transplanted mice (11.9/Litter) (Figure 2C). We examined the uterus of non-delivering recipients on day 22 of gestation, and found the Scd1^sgRNA^ embryos stopped development and were absorbed by the uterus (Figure 2B). From the genotyping analysis, we selected the mouse with 123 bp deletion in Exon1 and 1 bp insertion in Exon2 (Scd1^−/−^) and no off-target for the further experiments (Figure 2D,E and Appendix A–L). On day 5 after birth, the body length of Scd1^−/−^ mice (3.0 cm from the tip of the nose to the base of the tail) was significantly shorter than that of wild type (WT) mice (3.5 cm) (Figure 2F). We measured the bodyweight of mice from birth to 16 weeks of age and found that Scd1^−/−^ mice were significantly lighter than WT mice after the third week (Figure 2G). Furthermore, when compared to WT mice, Scd1^−/−^ mice lived shorter lifespan (Figure 2H). 

### 2.3. Scd1 Regulated the Blastocyst Generation and Lipid Droplet Synthesis in Mouse Embryos

The embryonic development rate for the 1- to 4-cell stage, the 4- to 8-cell stage, the morula, and the blastocyst in Scd1^−/−^ mice was compared to the WT at day 4.5 (Figure 3A and Table 2). Scd1^−/−^ embryos had a significantly lower blastocyst rate of 46.58% compared to WT (60.84%), and the majority of embryos were blocked at the morula stage (Figure 3D). The Scd1 protein level of those embryos was also decreased significantly (Figure 3B,C). Furthermore, the lipid droplet content was reduced in Scd1^−/−^ blastocysts (Figure 3E). To further demonstrate the role of Scd1 on embryonic development, we constructed the Scd1-overexpression vector (Appendix A) and examined the developmental rate (Appendix A) and lipid droplet content after overexpression (Appendix A). Scd1 over expression promoted the cleavage rate (Appendix A) and blastocyst rate (Appendix A) in both WT and Scd1^−/−^ embryos. Furthermore, after Scd1 overexpression, the lipid droplet content increased in both WT (Appendix A) and Scd1^−/−^ embryos (Figure 3E). Thus, Scd1-overexpression rescued the reductions in the cleavage rate and blastocyst rate, as well as lipid droplet content in embryos, which were caused by knockout of Scd1.

### 2.4. Single-Embryo RNA-Seq Atlas Exhibited the Increasing Requirement of Scd1 during Embryonic Development

To further investigate the specific molecular mechanisms underlying Scd1 participation in lipid synthesis during embryonic development, single-embryo transcriptome sequencing in zygote, 2-cell, 4-cell, and blastocyst stage embryos in both WT and Scd1^−/−^ embryos was performed and analyzed (Appendix A). *Scd1* gene expression started at the 2-cell stage and increased gradually as embryos developed, but it was significantly lower in *Scd1* knockout embryos than in WT embryos (Appendix A). In reference to the gene methylation database (http://bigd.big.ac.cn/methbank/org/, accessed on 18 July 2020), *Scd1* gene methylation level exhibited an opposite trend to its mRNA level (Appendix A). PCA analysis revealed that Scd1^−/−^ and WT embryos were clearly separated at each developmental stage (Appendix A).

### 2.5. Ribosome Biogenesis Was Suppressed in 2-Cell Stage Scd1^−/−^ Embryos

Through the analysis by the limma package, we obtained 2313 differentially expressed genes (DEGs) at the 2-cell stage, of which 1152 were up-regulated and 1161 were down-regulated (Appendix A). To further identify these DEGs, we selected 1553 DEGs concomitantly shown in the analyses by DESeq2, edgeR and limma (Appendix A). DEGs were enriched in the rRNA processing and maturation of the SSU-rRNA process (Appendix A). Gene Ontology (GO) and Kyoto Encyclopedia of Genes and Genomes (KEGG) analyses demonstrated the significant differential items, which mainly contained ribosome biogenesis, mRNA synthesis, and signaling pathways that regulate the pluripotency of stem cells (Appendix A). Gene Set Enrichment Analysis (GSEA) showed a more detailed functional enrichment: the ribosome biogenesis, peptide chain elongation, and 3′UTR mediated translational regulation processes were all inhibited in Scd1-knockout embryos (Figure 4A and Appendix A), whereas most of the ribosomal protein genes (RPs) were downregulated (Figure 4B), and genes related to drug metabolism cytochrome p450 and base excision repair were upregulated (Appendix A).

### 2.6. Ribosome Biogenesis and RNA Translation Was Reversely Upregulated in 4-Cell Stage Scd1^−/−^ Embryos

At the 4-cell stage, we identified 1316 DEGs by the limma package (Appendix A), and in combination with the DESeq2 and edgeR packages, 749 DEGs were concomitantly identified (Appendix A). These DEGs were enriched in the regulation of mRNA processing (Appendix A). GO and KEGG analyses demonstrated that the biological processes were enriched to microtubule and Golgi apparatus parts (Appendix A). GSEA analysis revealed a reverse regulation of processes such as RNA translation and ribosome biogenesis when compared to the 2-cell stage (Appendix A). Ribosome biogenesis and aging genes were upregulated (Appendix A), while the ERK-AMPK and TGF- pathways were downregulated (Appendix A). In brief, embryos exhibited a homeostasis-maintaining expression pattern during the 4-cell period, which was in contrast to the 2-cell period.

### 2.7. Ribosome Stress Stimulated the RPs-Mdm2-P53 Pathway in Scd1^−/−^ Blastocysts

At the blastocyst stage, 921 DEGs were identified by the limma package, with 459 genes up-regulated and 462 genes down-regulated (Appendix A). In combination with the analyses of the DESeq2 and edgeR packages, there were 359 DEGs that were concomitantly identified by the three packages (Appendix A). Notably, these DEGs were enriched in the P53 signaling pathway (upregulated) and cytosolic ribosome process (downregulated) (Figure 4C). GO and KEGG analyses also showed the same results. GSEA analysis showed that genes relating to the mTORC1, P53, and apoptotic pathway were upregulated, and genes relating to the rapamycin sensitive, ribosome and peptide chain elongation process were downregulated (Figure 4D and Appendix A). We intersected the genes in the P53 pathway and screened out 13 upregulated genes (Figure 4E,F), 11 of which were significantly different by *t*-test analysis (Figure 4E). Thus, following the inhibition of the ribosome biogenesis process during the 2-cell stage, *Scd1* gene knockout activated the P53 pathway in the blastocyst stage.

### 2.8. P53 Inhibition Rescued the Blastocyst Development in Scd1^−/−^ Embryos

Based on the above phenotypic studies and analysis of sequencing results, we further validated the P53 signaling pathway-mediated DNA damage repair marker γH2AX and reactive oxygen species (ROS) level during the blastocyst stage (Figure 5). The results showed that in Scd1^−/−^ embryos, the expression of P53 and γH2AX were upregulated (Figure 5A–C), the ROS content was also elevated (Figure 5E), and the expressions of the apoptosis-promoting gene *Bax* and genes relating to P53-mediated DNA damage repair (Figure 5D) and cell cycle regulation (Figure 5F,G) were all upregulated. Notably, PFT-α, a P53 pathway inhibitor, rescued the blastocyst rate caused by *Scd1* knockout (Figure 5H(a,b),I). However, Nutlin-3a, a P53 pathway activator, could significantly reduce the blastocyst rate in both WT and Scd1 overexpression embryos (Figure 5H(c,d),J). Thus, activation of the P53 pathway led to embryonic damage and oxidative stress, which arrested embryonic development, while inhibition of the P53 pathway could rescue the embryonic retardation.

### 2.9. Scd1^−/−^ Blastocyst Give Rise to Inner Cell Mess (ICM) Impairment and Embryo Development Arrest

To clarify the reason why blastocysts generation was arrested by *Scd1* knockout, we labelled trophectoderm (TE) cells with Cdx2 and ICM with Sox2 during the blastocyst stage (Figure 6). The results reflected a reduced proportion of ICM in Scd1^−/−^ blastocyst (Figure 6A), and the total cell number was also decreased significantly in Scd1^−/−^ blastocyst (Figure 6B). Then, in Scd1^−/−^ blastocysts, we counted the number of Cdx2 positive (Cdx2+) and Sox2 positive (Sox2+) cells compared to WT. In the blastocysts of 16–32 cells, 32–64 cells, and >64 cells, the percentages of Cdx2+ cells in Scd1^−/−^ embryos (79.95%, 76.88%, and 72.92%) were significantly higher than those in the WT embryos (57.37%, 61.71%, and 63.66%) (Figure 6C). In contrast, the ratio of Sox2+ cell in Scd1^−/−^ embryos (17.25%) was significantly lower than that in the WT embryos (34.30%) (Figure 6D,E). However, the Scd1-overexpression could rescue this result (Figure 5D). When Scd1^−/−^ embryos were compared to WT embryos, the embryonic stem cells (ESc) that separated from the ICM showed a differentiation trend (Figure 6F). As a result, hampered embryonic development caused by *Scd1* knockout may be due to ICM’s weakened stemness potential, which impairs cell division ability. When the embryo developed to day 12.5 (E12.5), there was a significant arrest shown in the Scd1^−/−^ embryos (Figure 6G,H), and the average litter size of Scd1^−/−^ mice (2.1 L) was significantly lower than that of WT mice (13.22/L) (Figure 6I).

## 3. Discussion

In this study, we created *Scd1* gene knockout embryos and a mouse model with a viability rate of 80–90% in embryos. Based on these models, we demonstrated that the *Scd1* gene is essential for the embryo development. Furthermore, using single embryo RNA sequencing, we identified a number of key factors that regulate embryonic development and clarified the regulatory mechanism in the embryo after *Scd1* knockout (Figure 7). For the first time, we reported that *Scd1* gene is not expressed at the initiation stage of zygotes until the embryo develops to the 2-cell stage. This may be due to the fact that at the 2-cell stage, the embryo begins to initiate zygote genome activation (ZGA) [33,34,35], and there is not much demand for the transcription of the *Scd1* gene.

As the embryo develops, lipids gradually become a source of energy supply [36,37], and *Scd1* expression increases correspondingly. If the lipid metabolism in the embryo is compromised due to the insufficient expression of the *Scd1* gene in Scd1^−/−^ mice, the embryos cannot develop healthily into the blastocyst [26,36,38]. The lack of energy also causes intra-embryonic damage and increased ROS production. In response to this stress, a number of ribosomal proteins bind to Mdm2 and prevent Mdm2-mediated ubiquitination of P53, leading to P53-dependent cell cycle arrest [39,40]. Ribosomal proteins play an important role in both disordered cell growth and cell division inhibition. In this way, the ribosomal protein-Mdm2-P53 signaling pathway provides a molecular switch that monitors the integrity of ribosomal biogenesis [41]. Panić et al. [42] reported that a reduction in individual RPs inhibited embryonic development after embryonic day 5.5 (E5.5) and caused apoptosis, and P53 deletion prolonged embryonic development to embryonic day 12.5 (E12.5). Thus, a P53-dependent checkpoint is activated during embryo development in response to ribosomal deficiency, preventing abnormal execution of the developmental program [43]. The P53 pathway is also activated to function as damage repair, causing cell cycle arrest or apoptosis during embryonic development. In vivo, inactivation of RPS6 activates a p53-dependent checkpoint [42,44], which is consistent with our findings in this study.

At the phenotypic level, knocking out the *Scd1* gene significantly reduces the blastocyst rate and triggers oxidative stress in the embryo, as well as P53-dependent cell cycle arrest. This may be due to the embryo development is an energy-intensive process. *Scd1*-knockout mediated lipogenesis triggers the intro-cellular stress and damage, which consistent with the viewpoint that inhibition of *Scd1* reduces fatty acid synthesis and increases β-oxidation, leading to a decrease in fatty acid stores [45,46]. It may also be due to the disruption of lipid metabolism, which generates ROS, which is byproduct of oxidative phosphorylation [47]. ROS also causes energy starvation by interfering with the function of proteins involved in energy metabolism [48]. Under an energy deficient stress, rRNA biogenesis is inhibited to conserve energy, while autophagy is induced to generate energy and maintain cell survival in the absence of energy [49]. Fatty acid metabolism provides energy for embryo development [36], and disruption of this process will inevitably affect embryo development. Our results in the present study showed that deletion of the *Scd1* gene not only reduced the total cell numbers of blastocysts but also destroyed the ICM formation, thereby reducing the blastocyst rate. However, the proportion of TE cells was compensatory enough to increase, which provided the embryo with the ability to repair the damage. 

Although 46.58% of Scd1-deficient embryos developed to the blastocyst stage, most of them were still lost in the growth process after transplantation, according to the present study. In addition, the body weight and subsequent survival rate of the mice were severely reduced. This study proved that the *Scd1* gene participated in the regulation of early embryonic development and was essential for the ICM generation. This is the first report on the role of the *Scd1* gene in reproduction in mice. However, the specific regulation mechanism needs further research to be confirmed.

In summary, the *Scd1* gene expression emerged when embryos began to initiate zygote genome activation, and knocking out the *Scd1* gene significantly reduced lipogenesis, triggered oxidative stress, activated the P53-dependent cell cycle arrest, and disrupted ICM formation in embryos, thereby preventing embryonic development. The results proved that *Scd1* gene-related lipid metabolism is essential to maintain healthy embryo development.

## 4. Materials and Methods

### 4.1. Cas9 mRNA Synthesis

For the production of Cas9 mRNA, the pST1374-NLS-flag-linker-Cas9 plasmid (Addgene#44758, Cambridge, MA, USA) was used, which was digested with Age I (R3552S, NEB, Ipswich, MA, USA) at 37 °C overnight. After the plasmid linearization, RNA secure reagent was added to the PCR reaction solution at a ratio of 1:25, and then the reaction solution was heated in a metal bath at 60 °C for 10 min. The linearized plasmid was purified with a PCR Purification Kit in an enzyme-free environment to remove the RNA enzyme, which was used as the template for Cas9 mRNA in vitro transcription (IVT) according to mMESSAGE mMACHINE™ T7 ULTRA (AM1345, Thermo Fisher Scientific, Rockford, IL, USA). After IVT, the Cas9 mRNA was obtained, and then it was purified using RNeasy Mini Kit. Finally, the quality of Cas9 mRNA was detected by 1% agarose gel electrophoresis. The Cas9 mRNA was stored at −80 °C. The digestion system was as follows: Age I, 8 μL; 10 × Cut Smart Buffer, 10 μL; p ST1374-NLS-flag-linker-Cas9, 8 μg; dd H_2_O, add to 100 μL.

### 4.2. sgRNA Synthesis

For the production of sgRNA, a linker sequence with the Bsa I restriction site was connected into the designed sgRNA, which upstream is TAGG and downstream is AAAC. After annealing, the synthesized oligonucleotide of the target site became double-stranded DNA, which connected with the pUC57 sgRNA expression vector (Addgene #51132, Cambridge, MA, USA). 

A single bacterial colony was picked and then sequenced to obtain a correctly constructed vector. The sgRNA in vitro transcription template was obtained by PCR from the pUC57 plasmid with IVT primers (Appendix A), and the product was purified using an Axygen PCR clean-up kit. Then, the transcription was performed using the MEGAshortscript^TM^ T7 Kit (AM1354, Thermo Fisher Scientific, Rockford, IL, USA). After transcription, DNase I was added to remove the template DNA, and the sgRNAs were purified with MEGAclear Kit (AM1908, Thermo Fisher Scientific, Rockford, IL, USA) according to manufacturer’s protocols. sgRNA target sites and oligonucleotides are available in Figure 1A and Table 1. The purified sgRNA was subjected to 180 V, 2% agarose gel electrophoresis to detect the quality. Then, the sgRNA was aliquoted and stored at −80 °C.

### 4.3. Animals

We created Scd1 knockout mice based on our previous research [32]. In addition, 8-week-old C57BL6 female mice and ICR mice were used as embryo donors and recipients.

The embryo-donor female mice were superovulated with 10 IU of pregnant mare serum gonadotropin (PMSG; Ningbo, China) first and 10 IU of human chorionic gonadotropin (hCG; Ningbo, China) 46–48 h later. Then they were mated with appropriate C57/BL6 male mice. After 14 to 16 h, the vaginal suppository was checked to prove successful mating. The mated mice were executed by a spinal dislocation method, and zygotes were collected from the ampulla of the fallopian tube. For recipient mice, on the day that donor mice were mating, they were mated with ligated male mice. Then, the zygotes underwent microinjection before being transplanted into the recipient’s fallopian tubes through the surgery method. After 21 days, those recipients would give birth to pups.

All animal experiments were performed in strict accordance with the Guide for the Care and Use of Laboratory Animals (Ministry of Science and Technology of the People’s Republic of China) and were approved by the Animal Care and Use Committee of Northwest A&F University.

### 4.4. Microinjection and Culture of Pre-Implantation Embryos

For microinjection, zygotes from superovulated and mated females were isolated 20 h post-hCG injection. Embryos were microinjected in M2 medium (M7167, Sigma, St. Louis, MO, USA) with a mixed sgRNA/Cas9 mRNA (25 ng μL^−1^/50 ng μL^−1^ or 50 ng μL^−1^/100 ng μL^−1^) into the nucleus between 27 h and 36 h after hCG injection, using an Eppendorf micromanipulator on a Zeiss inverted microscope. The injected embryos were cultured in EmbryoMax ^®^ KSOM Mouse Embryo Media (MR-121-D, Sigma, St. Louis, MO, USA) at 37 °C under 5% CO_2_ until the blastocyst stages.

### 4.5. Genotyping

Mice were genotyped using a 2 mm piece of the tail tip. The tail tips were incubated overnight at 55 °C in 400 μL of lysis buffer (50 mM NaCl, 10 mM Tris-HCl (pH 8.0), 5 mM EDTA, and 0.1% SDS) containing 150 μg of proteinase K. The next day, DNA was precipitated with an equal volume of isopropanol and dissolved in 200 μL TE buffer. DNA was extracted with the KAPA Express Extract Kit according to the manufacturer’s protocols.

For a single embryo, embryos at the blastocyst stage were collected and washed three times in PBS, and each of them was collected separately into an enzyme-free PCR tube. The DNA was extracted with the Qiagen REPLI-g Single Cell Kit (150343, Duesseldorf, Germany) according to the manufacturer’s protocols to amplify the embryo genome of mice. The genomic amplification product was diluted 10 or 20 times to a final DNA concentration of about 100 ng/L and used as a template to amplify each exon of the mouse *Scd1* gene with primers mE1 and mE2 (Appendix A) to obtain DNA fragments containing sgRNA targeting sites. The amplified DNA samples with double peaks were connected to the T vector. After transformation, 10 single clones were picked from each culture plate for sequencing and aligned with the wild-type gene sequence. PCR reactions were carried out using Prime STAR^®^ Max DNA Polymerase (Takara Bio Inc., Otsu, Japan). The predicted off-target sites were detected by PCR with the primers listed in Appendix A.

### 4.6. DNA Library Preparation for Single Embryo RNA-Seq

One embryo was collected in a PCR tube with the single cell collection solution that contains cell lysis components and RNase inhibitors. Reverse transcription was performed using nucleic acid sequences with oligo DT to form the 1st cDNA. The first cDNA was amplified by PCR to enrich nucleic acid, and the amplified product was purified for library construction, including DNA fragmentation, end repair, adding “a” and connector, PCR amplification and library quality control. The constructed library was sequenced by Illumina NovaSeq 6000. The sequencing strategy is PE150.

### 4.7. RNA-Seq Data Analysis

Raw sequences (raw reads) from Illumina platform sequencing were processed to obtain high-quality sequences (clean reads) by removing low-quality sequences, removing junction contamination, and so on. All subsequent analyses were based on clean reads. Then, the clean RNA-seq FASTQ data were mapped to the mouse reference genome GRCm39 using HISAT2 with the improved BWT algorithm [50] to convert the reference genome into the index. FPKM (Fragments per Kilobase per Million Mapped Fragments) is a very effective tool for quantitative estimation of gene expression values with a formula FPKM = 10^3^ * F/(NL/10^6^). F is the number of fragments uniquely matched to gene A, N is the total number of fragments uniquely matched to the reference gene, and L is the length of the exon region of gene A. The FPKM method can eliminate the influence of gene length and sequencing volume differences on the calculation of gene expression, and the calculated gene expression can be directly used to compare the gene expression differences among different samples.

For differential gene expression analysis, we used HTSeq with default settings to produce raw counts. With the raw count, differentially expressed gene analysis was performed by using the DESeq2, limma, and edgeR packages. Only genes with an adjusted *p* value less than 0.05 and at least a 2-fold change were considered to be differentially expressed. The GEO data were obtained from GSE18290 and GSE1749 [30,31].

### 4.8. Western Blot Analysis

In total, 120 embryos from each sample were collected in ice-cold lysis buffer for SDS-PAGE electrophoresis, and the semi-dry transfer system was used to transfer the protein to the FVDF membrane (HATF00010, Millipore, Burlington, MA, USA), which was blocked with 5% skim milk (232100, BD, Franklin Lakes, NJ, USA) at room temperature for 1.5 h. Then, the FVDF membrane was incubated overnight at 4 °C with the specific primary antibodies (P53: 60283-2-Ig, Proteintech Group, Wuhan, China; β-actin, CW0096, CW Biotech, Beijing, China). After washing three times with TBST, 10 min each time, the FVDF membrane was incubated with the secondary antibody for 1–1.5 h. Finally, the membrane was washed with TBST three times, 10 min each time, and detected with enhanced chemiluminescence (ECL) (1705061, Bio-Rad, Hercules, CA, USA). After incubating on the NC membrane, protein bands were detected and analyzed by the protein exposure system. β-Actin was used as an internal reference. 

### 4.9. RNA Extraction and Real-Time Quantitative PCR (RT-qPCR)

Total RNA was extracted from embryos at blastocyst stages using the Arcturus PicoPure RNA isolation kit and cDNA synthesis was carried out according to the protocols of Prime Script™ RT reagent Kit with gDNA Eraser (Perfect Real Time). RT-qPCR was operated according to the instructions of TB Green™ Premix Ex Taq™ II (RR820A, Perfect Real Time, Takara Bio Inc, Otsu, Japan). Relative levels of transcript expression were calculated using the 2^−∆∆Ct^ method with *Gapdh* as endogenous controls. The primers used were listed in Appendix A.

### 4.10. Immunofluorescence Staining

After removal of the zona pellucida with acidic Tyrode’s solution, mouse embryos were fixed in 4% PFA for 40 min at room temperature or overnight at 4 °C, followed by permeabilization in 0.5% Triton X-100 for 30 min. Embryos were blocked at 37 °C for 2 h or at 4 °C in 3% BSA in PBS-T (PBS containing 0.1% Tween) overnight and then incubated with primary antibodies (Scd1: mAb #2794, Cell Signaling Technology, Danvers, MA, USA; γH2AX: ab81299, Abcam, Cambridge, UK; Sox2: sc-365823, Santa Cruz Biotechnology, USA; Cdx2: BM4014, BOSTER, Wuhan, China) in the blocking solution overnight at 4 °C. The embryos were then washed twice with PBS-T and incubated with the secondary antibodies for 2 h before final washes in PBS-T and imaging in drops of PBS on glass-bottomed dishes, covered by paraffin mineral oil.

### 4.11. Lipid Droplet Staining

BODIPY493/503 (D3922, Invitrogen, USA) was used for the lipid droplet staining in embryos. Mouse embryos were fixed in 4% PFA for 40 min at room temperature or overnight at 4 °C, followed by permeabilization in 0.5% Triton X-100 for 30 min. After three times of washing in DPBS, the embryos were incubated in the BODIPY (1μg/mL in DPBS) dyestuff for 30 min at room temperature in a dark environment and the nuclei were incubated with DAPI (C0065, Solarbio, Beijing, China) for 10 min. Then, the fluorescence was detected under the microscope (EVOS M5000, Thermo Fisher Scientific, Waltham, MA, USA).

### 4.12. Statistical Analysis

Statistical analyses were performed with the SPSS 18.0 statistics software or package Prism 9 (GraphPad). Data are presented as means ± SD (standard deviation) of three independent experiments. Significant differences between different groups were determined using Student’s unpaired *t*-tests, taking * *p* < 0.05, ** *p* < 0.01 as significant differences, ns as not significant.

## 5. Conclusions

In conclusion, the *Scd1* knockout arrested the mouse embryo development, resulting in a lower blastocyst rate and smaller litter size. The effects were mediated by lipid droplet content and the RPs-Mdm2-P53 pathway, which activated apoptosis genes and caused ICM stemness potential to be lost. The effects of the *Scd1* gene in blastocyte implantation, embryo development, and fetus growth cannot be ruled out and need to be investigated. In addition, studies on the effect of this gene on embryonic pluripotency will be carried out subsequently.

## Figures and Tables

**Figure 1 ijms-24-01750-f001:**
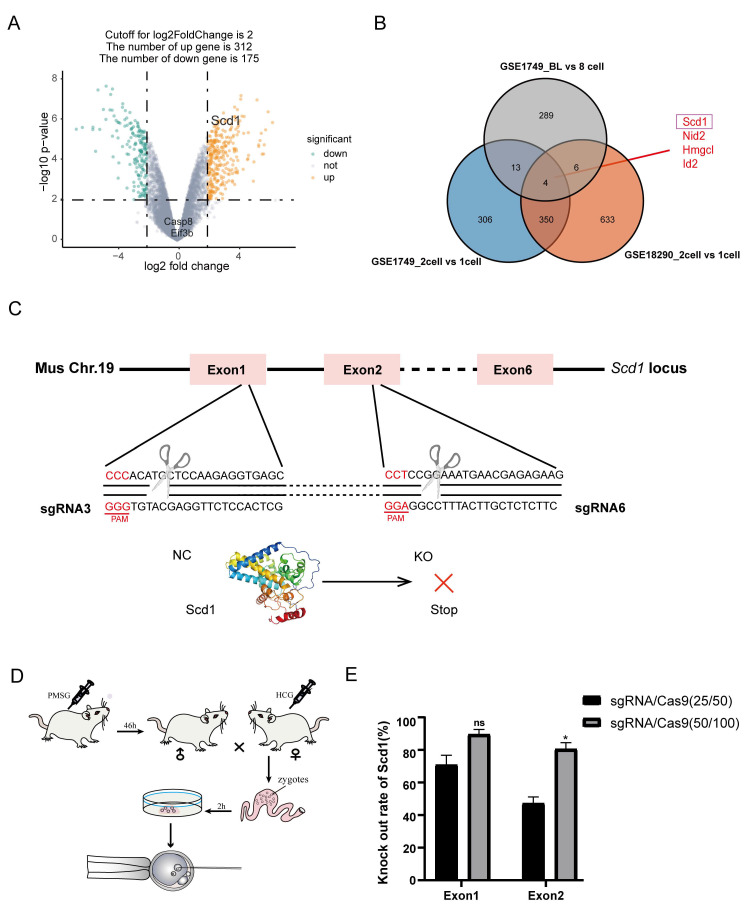
Expression of *Scd1* gradually increases as embryonic development. (**A**): The Volcano map of differentially expressed genes (DEG) of GSE1749, a data set characterizing embryonic development. (**B**): Venn diagram showing genes co-upregulated in 2-cell and blastocyst stages in GSE1749 and GSE18290. (**C**): The target sites of sgRNA3 and sgRNA6 on *Scd1* gene locus. (**D**): Schematic diagram of embryo collection and microinjection. (**E**): Comparation of *Scd1* gene knock out efficiency between two different concentrations of sgRNA/Cas9 (sgRNA:25 ng/μL, Cas9:50 ng/μL; sgRNA:50 g/μL, Cas9:100 ng/μL) at exon1 and exon2. Two-tailed Student’s *t*-tests were used for statistical analysis, * *p* < 0.05; ns, not significant (*p* > 0.05).

**Figure 2 ijms-24-01750-f002:**
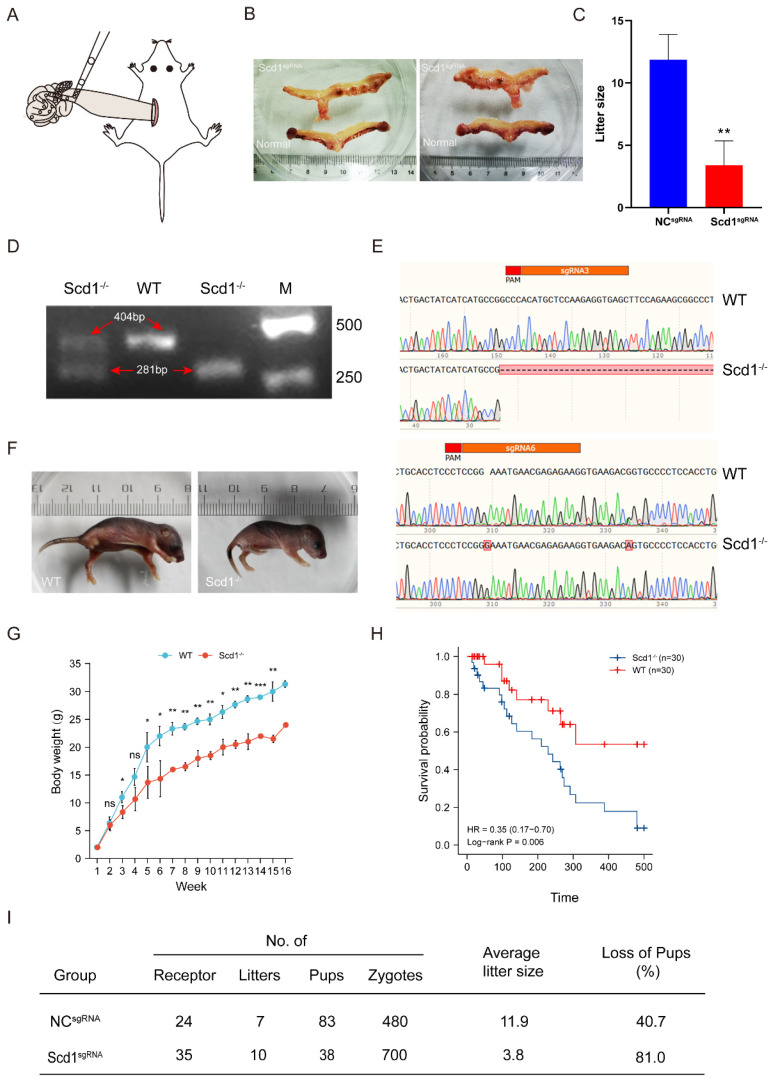
Scd1 deficiency causes embryonic development arrest. (**A**): The diagram of embryo transplantation. (**B**): The uterus of not delivered mice that was transplanted with Scd1^sgRNA^ embryos at day 22 of gestation. (**C**): The litter size of mice transplanted with NC^sgRNA^ and Scd1^sgRNA^ embryos. (**D**): Gel analysis of wild type (WT) and *Scd1* knockout mice. (**E**): The gene sequences of the WT and Scd1^−/−^ mice which has 123 bp deletion on the Exon1 and 1 bp insertion on the Exon2. (**F**): Body size of WT and Scd1^−/−^ mice at day 5 after born. (**G**): Body weight of mice for 16 consecutive weeks in WT and Scd1^−/−^ group. (**H**): The comparation of Survival curve of Scd1^−/−^ mice and WT mice. The Log-rank test was used for the statistical analysis. *p* = 0.006. (**I**): The data of embryo loss after transplantation. * *p* < 0.05; ** *p* < 0.01; *** *p* < 0.001, ns: not significant.

**Figure 3 ijms-24-01750-f003:**
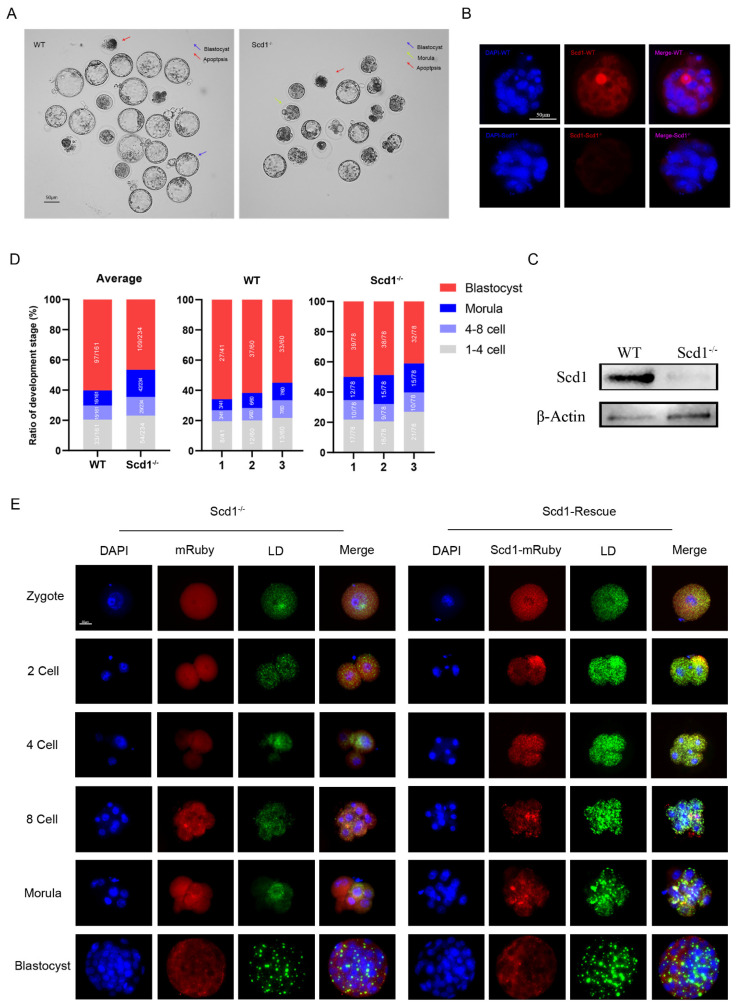
Scd1 regulates the blastocyst generation and lipid droplet synthesis in mouse embryos. (**A**): The comparation of blastocyst development rate between the WT and Scd1^−/−^ groups. Bar, 50 μm. (**B**): Immunofluorescence staining of Scd1 in the WT and Scd1^−/−^ groups, Bar, 50 μm. (**C**): the western blot detection of Scd1 in the WT and Scd1^−/−^ groups. (**D**): Statistics of embryo development in two groups. Data represent means of all embryos acquired from three independent culture units (1: indicates unit 1; 2: indicates unit 2 and 3: indicates unit 3). (**E**): Bodipy staining (green: refers to the lipid droplet) of Scd1^−/−^ embryos with or without Scd1-mRuby (rad) rescue. Bar, 30 μm.

**Figure 4 ijms-24-01750-f004:**
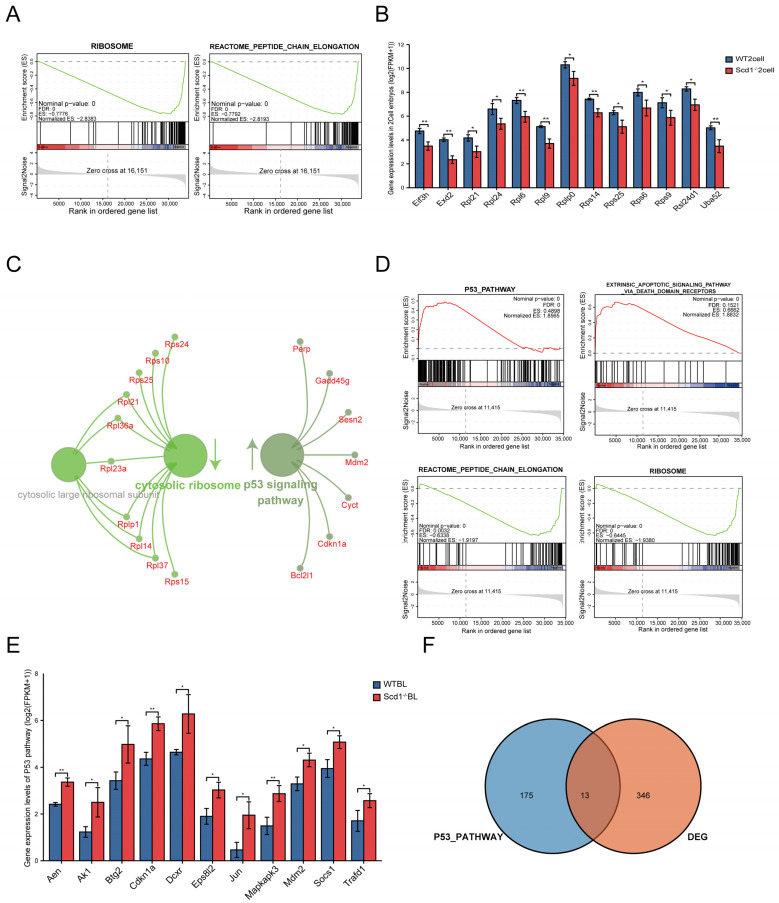
Scd1-knockout leads to ribosome stress and stimulates the RPs-Mdm2-P53 pathway. (**A**): The GSEA result for the Scd1^−/−^ versus WT embryos at 2-cell stage. (**B**): Genes related to ribosome biogenesis process were downregulated significantly in Scd1^−/−^ 2 cell stage embryos. (**C**): Functional enrichment analysis of differential genes between Scd1^−/−^ and WT embryos at blastocyst stage. (**D**): The GSEA result for the two groups’ embryos in blastocyst stage. (**E**): The expression levels of DEGs related to P53 pathway at blastocyst stage. * *p* < 0.05; ** *p* < 0.01. (**F**): Venn diagram of 13 genes in both DEGs and P53 pathway at blastocyst stage.

**Figure 5 ijms-24-01750-f005:**
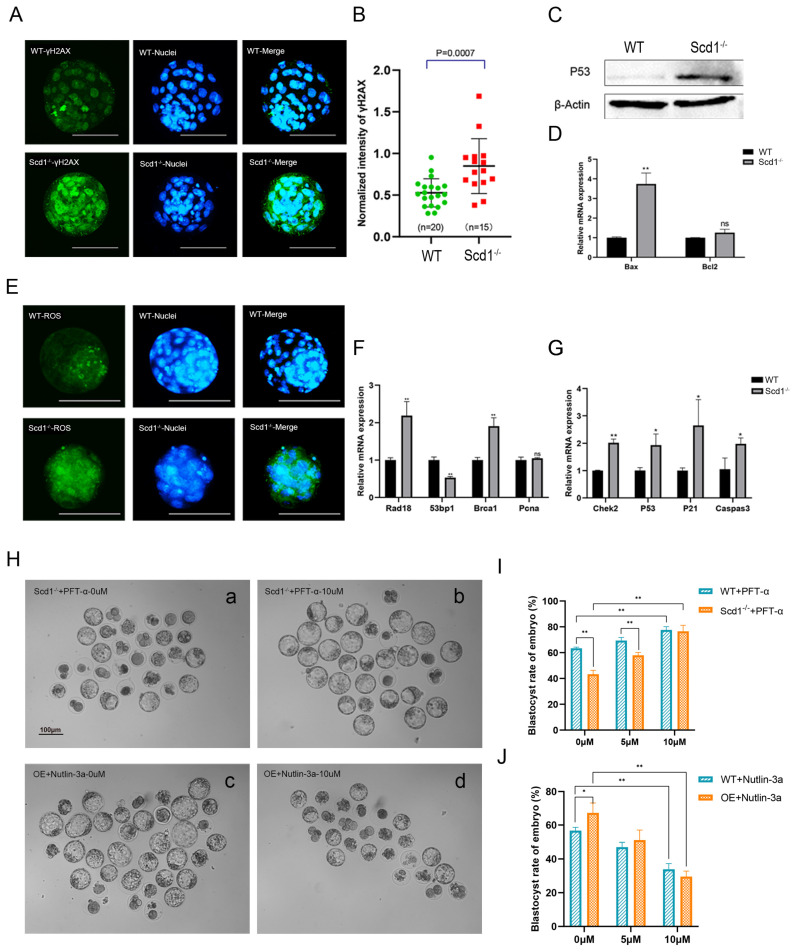
P53 inhibition rescues the blastocyst development in Scd1^−/−^ embryos. (**A**): Immunofluorescence assay of γ-H2AX level in WT and Scd1^−/−^ blastocysts. Scale bars, 75 μm. (**B**): Immunofluorescence intensity analysis. n = number of embryos. (**C**): Detection of P53 protein expression using Western blot. (**D**): Expression level of apoptosis relative genes (Bax and bcl2). (**E**): Detection of reactive oxygen species (ROS) in WT and Scd1^−/−^ blastocysts. Scale bars, 100 μm. (**F**): Relative gene expressions of DNA Post-replication modification and P53-pathway (**G**) in WT and Scd1^−/−^ blastocysts. * *p* < 0.05; ** *p* < 0.01, (*t*-test). (**H**): (**a**,**b**): The blastocyst development rate of Scd1^−/−^ embryos with or without P53 pathway inhibitor PFT-α (10 μM) treatment. (**c**,**d**): The blastocyst development rate of embryos after Scd1 overexpression (OE) with or without P53 pathway activator Nutlin-3a (10 μM) treatment. Bar, 100 μm. (**I**): The statistical analysis of (**H**) (**a**,**b**) and (**J**): (**H**) (**c**,**d**). The concentration of PFT-α and Nutlin-3a was 0 μM, 5 μM, 10 μM.

**Figure 6 ijms-24-01750-f006:**
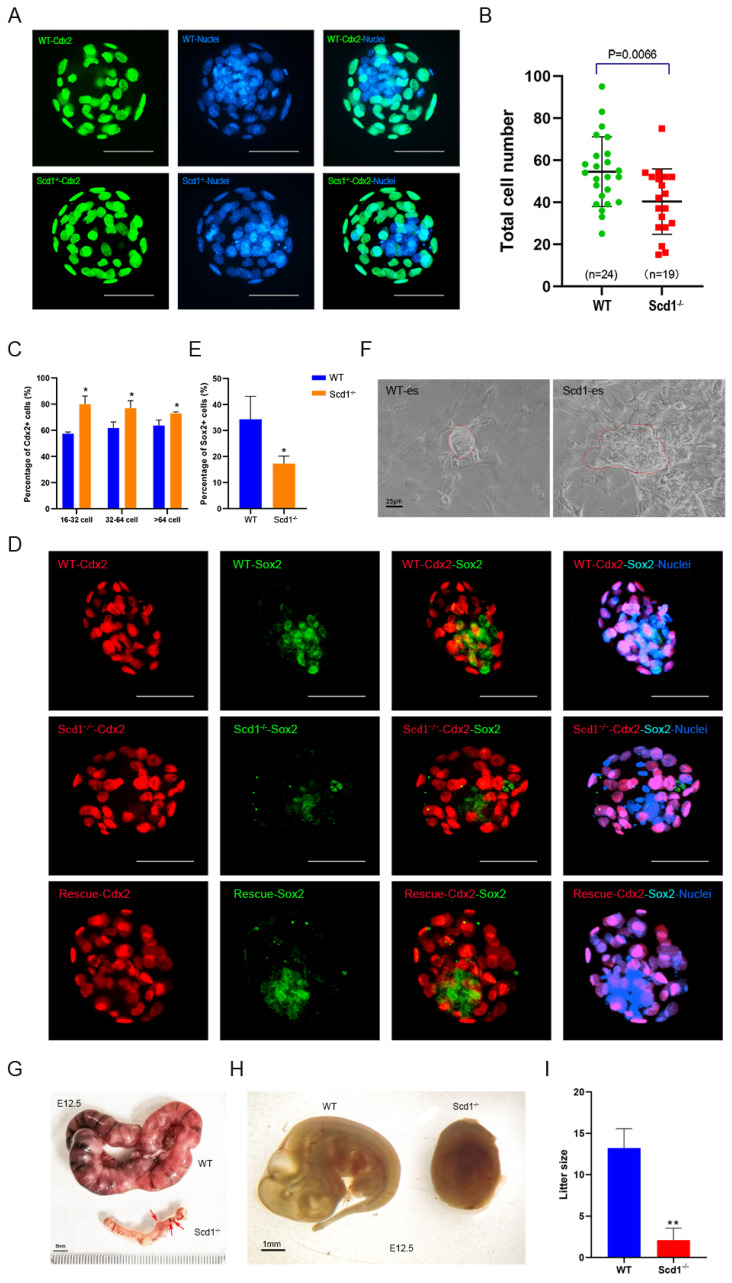
Scd1^−/−^ blastocyst give rise to inner cell mess (ICM) impairment and embryo development arrest. (**A**): Immunofluorescence staining of Cdx2 (Trophectoderm cell marker) in WT and Scd1^−/−^ blastocysts. Scale bars, 50 μm. (**B**): The total cell number of WT and Scd1^−/−^ blastocysts. Two-tailed Student’s *t*-test was used for statistical analysis. The Scd1^−/−^ group has significantly fewer cell numbers than the WT group (*p* = 0.0066). Each dot represents one embryo, and black bars indicate the mean cell number for each group. (**C**): Number of Cdx2^+^ cells in Scd1^−/−^ blastocysts compared with WT in 16–32 cells, 32–64 cells and >64 cells embryos, respectively. * *p* < 0.05 (*t*-test). (**D**): Immunofluorescence staining of Sox2 (inner cell mass marker) and Cdx2 in WT and Scd1^−/−^ blastocysts. Rescue group represent the overexpression of Scd1 in Scd1^−/−^ blastocysts to reflect the rescue function of Scd1 in ICM generation. Scale bars, 50 μm. (**E**): Numbers of Sox2^+^ cells in WT and Scd1^−/−^ blastocysts. * *p* < 0.05 (*t*-test). (**F**): Morphology of the embryo stem (es) cells isolated from the ICM of WT and Scd1^−/−^ blastocysts. Bar, 25 μm. (**G**): The uterus of WT and Scd1^−/−^ mice when E12.5 of pregnancy. The red arrow marked the embryos’ location. Bar, 5 mm. (**H**): The embryos (E12.5) of WT and Scd1^−/−^ mice. Bar, 1 mm. (**I**): The litter size of mice in two groups. ** *p* < 0.01 (*t*-test).

**Figure 7 ijms-24-01750-f007:**
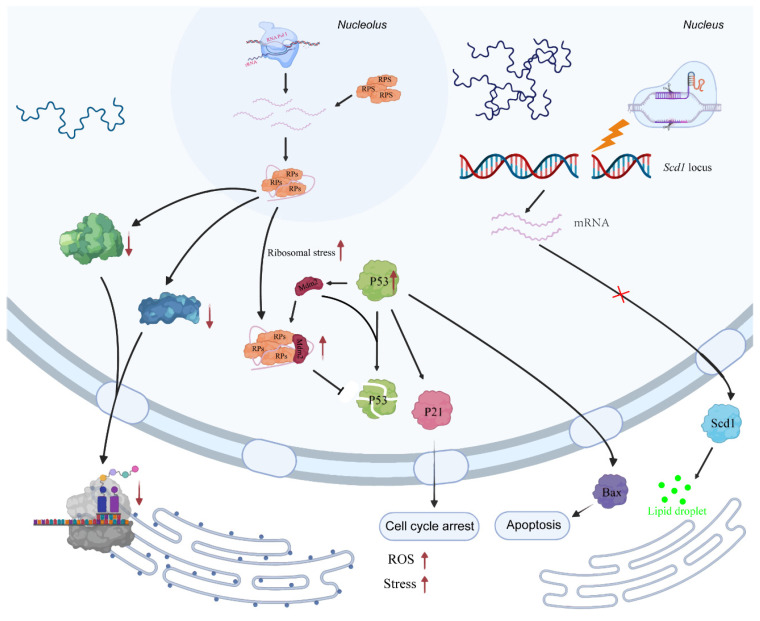
Overview of the molecular processes occurring after *Scd1* knock out. A schematic illustration showing that knockout of *Scd1* gene triggered insufficient of lipid droplet. Then the RPs-Mdm2-P53 pathway was activated, and genes related to apoptosis and cell cycle arrest were upregulated, leading to the increasing of ROS and ribosome stress, and inhibiting of ribosome biogenesis and embryonic developmental arrest. RPs: r-protein, including RPL and RPS, which bind to rRNA and form the large and small subunits of the ribosome, respectively.

**Table 1 ijms-24-01750-t001:** sgRNA sequences target to mouse *Scd1* gene.

sgRNA	Sequence (Bases in Bold Are PAM Sequence)
sgRNA3	GCTCACCTCTTGGAGCATGT**GGG**
sgRNA6	CTTCTCTCGTTCATTTCCGG**AGG**
sgRNA-NC	ACCGGAAGAGCGACCTCTTCT

sgRNA3 and sgRNA6 were target to the first and second Exon region of *Scd1* gene, respectively. sgRNA-NC is an RNA sequence without any target site.

**Table 2 ijms-24-01750-t002:** The effect of *Scd1* knockout on blastocyst development rate.

Group	Total Number (tn)	Morula Number (mn)	Morula Rate (%)	Blastocyst Number (bn)	Blastocyst Rate (%)
Scd1^−/−^	234	42	17.95 ± 1.81	109	46.58 ± 3.96 ^a^
WT	161	16	9.66 ± 1.79	97	60.84 ± 4.47 ^b^

Total number (tn) represent the sum of embryo observed at blastocyst stage. Morula number (mn) and Blastocyst number (bn) refer to the embryos at these stages. $\mathrm{Morula}\;\mathrm{rate}=\frac{\mathrm{mn}}{\mathrm{tn}}\times 100\%$. $\mathrm{Blastocyst}\;\mathrm{rate}=\frac{\mathrm{bn}}{\mathrm{tn}}\times 100\%.$ Data are obtained from at least three repeat tests, and are represented as mean ± SEM. ^a,b^ different lowercase letters represent significant difference (*p* < 0.05).

## Data Availability

The data that support the findings of this study are available from the corresponding author upon reasonable request.

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
