# Peer review of "Scd1 Deficiency in Early Embryos Affects Blastocyst ICM Formation through RPs-Mdm2-p53 Pathway"

_ijms, 2023, doi:10.3390/ijms24021750_

Round 1

Reviewer 1 Report

I have some minor suggestions and comments:

1.    In paragraph 2.9 (lines 310-328) there is no reference to Figure 6I.

2.    The publication cited in subsection 4.3 Animals in line 440 at [30] should be [47].

3.    In the References section, page numbers are missing or incorrectly stated in the following reference numbers: 2, 10,12,16,24,26,32,35,37,39,40.

E.g. Publication No. 2 is: 593-7 and should be 593-597;

Publication No. 16 is missing: 39(12), 1660-1670, …… etc.

Author Response

Response to Reviewer 1 Comments

Point 1: In paragraph 2.9 (lines 310-328) there is no reference to Figure 6I.

Response 1: We are grateful for your professional review of our article. As you are concerned, Figure 6I has been added in paragraph 2.9 (lines 328).

Point 2: The publication cited in subsection 4.3 Animals in line 440 at [30] should be [47].

Response 2: Thank you for your suggestion. We are sorry that we did not make it clear that the reference [30] is one of our previous studies and that we generated the knockout mice based on the same method. So, we cited it here.

Point 3: In the References section, page numbers are missing or incorrectly stated in the following reference numbers: 2, 10,12,16,24,26,32,35,37,39,40.

Response 3: Thank you for your reference-specific queries. The format of the references you mentioned has been corrected.

Reviewer 2 Report

The present manuscript studied SCD1’s function during embryogenesis. The authors identified SCD1 was one of those unregulated genes compared 2-cell vs 1-cell stage, and BL vs 8 cell stage using public data. Using CRISPR/Cas9 knocking out of SCD1 in embryos impaired the development of embryos, which confirmed the importance of SCD1. RNAseq analysis of the gene expression pattern of WT and SCD1 KO embryo at different stages, identified P53 pathway involved in SCD1 deficiency caused embryonic development arrest. The approach to reach such conclusion is well thought and smart. However, there are obvious over-statements. The second major weakness is language and figure organization, which will confuse readers and make it not easy to read. The authors should consider the following points.

1. The authors should describe results subjectively, and make statement based on the current data. For example, in the title and abstract, the authors mentioned RPs-Mdm2-p53 pathway, however, in this manuscript, there’s no experimental data about Mdm2; in the abstract, the author also claimed SCD1’s function is providing energy for embryonic development, there’s no data to prove it in this manuscript.

 2. it is well known SCD1 is mainly to produce MUFAs, the authors should add back MUFAs to SCD1 KO embryos to test if it could rescue the development arrest phenotype.

3. Page 2 line 83, GSE1794 should be 1749, and the authors should reference these 2 sets of data they used.

4. Figure 2E, better show include WT, SCD1 KO, SCD1 KO + OE,  merge lane is not informative may be removed. Figure S2 C,D, should compare WT vs SCD1-OE not SCD1 KO vs SCD1-OE.

5. Figure 5, H(cd) and J better come earlier than H(ab) and I, also I need statistics.

6. Figure 6, WT first, then SCD1 KO.    

7. There’s no Figure 7.

8. There are too many typos, need notice space between words.

Author Response

Response to Reviewer 2 Comments

Point 1: The authors should describe results subjectively, and make statement based on the current data. For example, in the title and abstract, the authors mentioned RPs-Mdm2-p53 pathway, however, in this manuscript, there’s no experimental data about Mdm2; in the abstract, the author also claimed SCD1’s function is providing energy for embryonic development, there’s no data to prove it in this manuscript.

Response 1: We feel great thanks for your professional review work on our article. As you are concerned, RPs-Mdm2-p53 pathway was shown important in embryo development, the expression changes of RPs and Mdm2 were found and clarified in the RNA-Seq result (Fig.6 B and E) during the 2cell and blastocyst stage. Besides that, many researches have reported the relationship of RPs, Mdm2 and P53 (reference 38-40). For Scd1’s function in providing energy, we detected the count of LDs in WT and Scd1-/- embryos, which represent the energy in the form of fat. For other forms of energy detection, we will complete in subsequent experiments. When completed, we will also submit to your journal.

Point 2:  it is well known SCD1 is mainly to produce MUFAs, the authors should add back MUFAs to SCD1 KO embryos to test if it could rescue the development arrest phenotype.

Response 2: Thank you for your suggestions. Your comment is of great importance to our article. It has contributed a lot to improving the quality of our article. The effect of adding MUFAs on embryonic development will be completed in a follow-up trial and will continue to be submitted to your journal upon completion.

Point 3: Page 2 line 83, GSE1794 should be 1749, and the authors should reference these 2 sets of data they used.

Response 3: Thank you for your valuable comment. We are very sorry for our incorrect writing of GSE1794, we have made a correction in red color. The 2 sets of data were referenced in the section 2.1 (line 83 and 84) and Materials and Methods section 4.7 (line 506-507).

Point 4: Figure 2E, better show include WT, SCD1 KO, SCD1 KO + OE, merge lane is not informative may be removed. Figure S2 C,D, should compare WT vs SCD1-OE not SCD1 KO vs SCD1-OE.

Response 4: Thank you for your kind suggestions. Figure 2E showed the sequencing results of the Scd1 knockout mice. So, what you mentioned is perhaps Figure 3E. The merge lane appears here with the intention of showing whether Scd1 expression and LDs have co-localization. Indeed, the enhanced signal of Scd1 localization was consistent with LDs in the two-cell stage Scd1-OE embryos. The figure S2 C,D, compared the cleavage rate and blastocyst rate of four groups (WT, Scd1-/-, Scd1-OE, Scd1-/--OE). And the figure S2 E,F, compared the LDs content of WT vs SCD1-OE.

Point 5: Figure 5, H(cd) and J better come earlier than H(ab) and I, also I need statistics.

Response 5: Thank you for your recommendation. The order of H(cd) and H(ab) has been noted and ordered in Figure 5, as well as the article's corresponding place. The statistical results have been reflected in I and J.

Point 6: Figure 6, WT first, then SCD1 KO.

Response 6: Thank you for your corrections. The order of WT and KO has been changed in Figure 6.

Point 7: There’s no Figure 7.

Response 7: Thank you for your suggestion. We are very sorry for our negligence of missing Figure 7, and it has been added in page 20.

Point 8: There are too many typos, need notice space between words.

Response 8: The revision is reviewed by expert and possible corrections were made to eliminate typo errors.